# Nanobodies Provide Insight into the Molecular Mechanisms of the Complement Cascade and Offer New Therapeutic Strategies

**DOI:** 10.3390/biom11020298

**Published:** 2021-02-17

**Authors:** Alessandra Zarantonello, Henrik Pedersen, Nick S. Laursen, Gregers R. Andersen

**Affiliations:** 1Department of Molecular Biology and Genetics, Aarhus University, 8000 Aarhus, Denmark; azara@mbg.au.dk (A.Z.); henrik@mbg.au.dk (H.P.); 2Department of Biomedicine, Aarhus University, 8000 Aarhus, Denmark; nsl@biomed.au.dk

**Keywords:** complement system, proteolytic cascade, convertase, inhibitor, structural biology, molecular mechanism

## Abstract

The complement system is part of the innate immune response, where it provides immediate protection from infectious agents and plays a fundamental role in homeostasis. Complement dysregulation occurs in several diseases, where the tightly regulated proteolytic cascade turns offensive. Prominent examples are atypical hemolytic uremic syndrome, paroxysmal nocturnal hemoglobinuria and Alzheimer’s disease. Therapeutic intervention targeting complement activation may allow treatment of such debilitating diseases. In this review, we describe a panel of complement targeting nanobodies that allow modulation at different steps of the proteolytic cascade, from the activation of the C1 complex in the classical pathway to formation of the C5 convertase in the terminal pathway. Thorough structural and functional characterization has provided a deep mechanistic understanding of the mode of inhibition for each of the nanobodies. These complement specific nanobodies are novel powerful probes for basic research and offer new opportunities for in vivo complement modulation.

## 1. Introduction 

### 1.1. The Complement System

In the following, we describe those pathways and molecules from the complement system that are needed to understand the experimental results described in the upcoming sections. For more comprehensive reviews of specific aspects, the reader is referred to recent review works including [1,2,3].The complement system is an efficient weapon of innate immunity, which opsonizes the surface of invading organisms and apoptotic host cells for elimination through phagocytosis and cell lysis. In the innate immune system, pattern recognition molecules (PRMs) bind pathogen-associated molecular patterns and damage-associated molecular patterns [3,4]. Complement activation also elicits an inflammatory response at the site of infection. The complement cascade can be activated through three distinct pathways; the classical pathway (CP), the lectin pathway (LP), and the alternative pathway (AP) (Figure 1). Both the classical and the lectin pathways initiate by activation of giant complexes formed between an oligomeric pattern recognition molecule and a protease complex that cleaves complement component C4 resulting in deposition of the major fragment C4b on the activator (Figure 1). In the CP, C1q is the pattern recognition molecule and circulates in complex with the serine proteases C1r and C1s (C1 in Figure 1). C1q is a hexamer of trimers, and each trimer is formed by chains A, B and C. The C-terminal parts of the three chains fold into structural entities known as the C1q globular domains, while the N-terminal parts of the three chains are joined in a collagen helix. The C1 complex initiates the CP when C1q recognizes antigen bound immunoglobulin G (IgG) and immunoglobulin M (IgM) (Figure 1), the acute phase proteins CRP, pentraxins and anionic phospholipids as phosphatidylserine on apoptotic and necrotic cell surfaces [5,6]. C1q similarly recognizes LPS in the cell wall of Gram-negative bacteria as well as viral proteins [7]. In the brain, C1q binds Aβ and PrP oligomers, playing a role in Alzheimer’s and prion disease progression, respectively [8,9]. The vast majority of C1q binding patterns are recognized by the C1q globular domain. Specifically, the C1q B and C chains establish direct interactions with the fragment crystallizable (Fc) moiety of IgG and IgM in immune complexes [3,5,10,11,12,13,14]. Activated C1s in the C1 complex cleaves C4 to C4b, and a major conformational change in nascent C4b exposes a reactive thioester group, which may react with a nucleophile leading to covalent attachment of C4b to the activator (Figure 1) [15,16]. The C4b conformation allows binding of the zymogen C2, to form the proconvertase C4b2. Within the proconvertase, C2 is cleaved by C1s resulting in formation of the C3 convertase C4b2a [17,18,19]. In the related LP, the cascade can be initiated by five different PRMs called mannan-binding lectin (MBL), M-, L-, H-ficolins (or ficolin-1 to 3) and CL-LK [20,21,22] upon binding to conserved carbohydrate structures on the activator surface. The LP PRMs circulate in complex with dimers of the MBL-associated serine proteases (MASPs) 1, 2, and 3 [23,24,25]. MASP-1 and MASP-2 are functional homologs of C1r and C1s in the CP, and upon activation of the LP cascade, C4 is cleaved and the same proconvertase and C3 convertase are assembled as in the CP. Since the C4b2a C3 convertase is the endpoint of both pathways, it is referred to as the CP/LP C3 convertase. C4b2a cleaves its substrate C3, which undergoes a conformational change similar to nascent C4b, and exposes the reactive thioester that forms an ester bond with a hydroxyl nucleophile on the surface (Figure 1) [26,27]. 

In the alternative pathway, C3b deposited by the CP/LP C3 convertase binds to factor B (FB) to form the AP proconvertase C3bB (Figure 1). The fluid phase protease Factor D (FD) cleaves FB in the proconvertase and the AP C3 convertase C3bBb appears [28,29,30]. The ability of C3b to form a C3 convertase and promote further C3 cleavage gives rise to the AP amplification loop, which amplifies the C3b deposition catalyzed by the CP/LP C3 convertase by 5-10 fold [31,32,33]. Besides starting from C3b deposited through the CP and LP, a fluid phase AP C3 convertase may assemble after spontaneous C3 hydrolysis (tickover mechanism) whereby a C3b-like molecule called C3(H_2_O) is formed that is capable of binding to FB [34]. This fluid phase proconvertase is also activated by FD cleavage, however the physiological role of the fluid phase AP C3 convertase is still debated [35,36]. The half-life of the AP C3 convertase is only 90 s under physiological conditions, but it is increased up to ten-fold by binding of properdin (FP), the only known positive regulator of the complement system [37,38]. FP circulates as dimeric, trimeric, or tetrameric homo-oligomers and recognizes primarily C3b within the AP C3 convertase [39].

Both C4b and C3b are tightly regulated to avoid complement activation on host cells, but due to the strong amplification through the alternative pathway, C3b regulation is likely to be the most important in vivo. Regulators of complement activity (RCA) expressed on or recruited to host cells may exert decay acceleration by dissociating the C3 convertases, but may also act as cofactors for C4b and C3b degradation by the protease factor I (FI) [40] (Figure 1). The products of FI degradation, C4c C4d, iC3b and C3dg are inactive with respect to convertase formation. The best characterized FI cofactors are factor H (FH), membrane cofactor protein (MCP/CD46) and CR1 (CD35). MCP and CR1 are membrane bound and recognize both C4b and C3b [41,42]. In contrast, FH is C3b specific and acts both in the fluid phase and on host cells through recognition of sialic acid and glycosaminoglycan [43,44]. On non-host cell like pathogens, conversion of C3b to iC3b is slower than on host cells due to weaker binding of FH and the lack of membrane bound regulators.

The multiple effector functions elicited by the C3 degradation products are reviewed in [45]. C3 cleavage releases the anaphylatoxin C3a, which triggers an inflammatory response at the activation site through binding to the G-protein coupled receptor C3aR [46]. Another effector function of C3 fragments is to induce phagocytosis of the activator. C3b and iC3b mediate phagocytosis through interaction with complement receptor Vsig4 (also called CRIg) presented by Kupffer cell macrophages [47,48], while only iC3b are recognized by CR3 and CR4 to elicit phagocytosis. CR3 and CR4 are expressed in the myeloid subsets of leukocytes, on NK cells and activated T and B lymphocytes [49]. Furthermore, iC3b and its degradation product C3dg confer cross talk between the complement cascade and adaptive immunity through their binding to CR2 on B lymphocytes [50].

If not degraded by FI, the surface density of C3b continues to increase on the activator. When a threshold density is reached, both C3 convertases shift their substrate specificity from C3 to C5 [51,52]. C5 is structurally homologous to complement C3 [53], but does not have an internal thioester. Instead, C5b interacts with C6, forming the C5b6 complex, which can transiently associate with a nearby lipid bilayer on a complement opsonized cell. Subsequent recruitment of C7 leads to stable association with the membrane, and lipid bilayer penetration starts when C8 joins the complex. The pore size is increased by insertion of several C9 molecules, resulting in formation of the membrane attack complex (MAC) that perforates the cell (Figure 1) [54]. The lytic ability of complement is important for killing of Gram negative bacteria, one important example is *Neisseria meningitidis* [55]. The cleavage of C5 also leads to the release of C5a, which triggers a potent inflammatory response through binding to the G-protein coupled receptor C5aR1. This induces vasodilation, release of histamine and contraction of smooth muscle, as well as chemotaxis of neutrophils, T cells, activated B cells, macrophages and basophils [56].

### 1.2. The Complement System as a Driver of Pathogenesis

Regulators of complement are ubiquitously expressed on host surfaces to control undesired amplification, and tipping of this balance is a major mechanism for diseases associated with complement activation. It is well established that especially aberrant AP activity on self surfaces is associated with the development of atypical hemolytic uremic syndrome (aHUS), paroxysmal nocturnal hemoglobinuria (PNH), age related macular degeneration (AMD), anti-neutrophilic cytoplasmic autoantibodies (ANCA) vasculitis, and C3 glomerulopathy (C3G) [45]. PNH arises from clonal expansion of hematopoietic stem cells that comprise a loss-of-function mutation in the *PIGA* gene [57] that is essential for synthesis of glycosylphosphatidylinositol (GPI) anchors and the mutation hence results in deficiency of GPI anchored proteins, including DAF [58] and CD59 [59]. Lack of these surface bound complement regulators render erythrocytes vulnerable to complement attack and lysis by the membrane attack complex. The clinical manifestations of the disease include thrombosis and anemia [57]. Similarly, complement exacerbates the development of aHUS that arises from dysregulation of complement on host endothelia, most commonly the kidneys [60]. Patients suffering from aHUS often possess mutations in FH [61] which may result in reduced levels of surface-bound FH and consequently reduced protection against complement activation. Similarly, genetic studies report loss-of-function mutations in genes encoding the complement regulators FI and MCP as well as gain-of-function mutations in genes for FB and C3 in aHUS patients [62]. The term ‘C3 glomerulopathy’ is used to describe glomerular disorders, where complement dysregulation either underlies or exacerbates disease development [63]. A common characteristic of C3 glomerulopathies (C3G) is C3 fragment deposition in the renal tissue leading to irreversible kidney damage. C3G arise from dysregulation of the AP in fluid phase driven by either acquired or genetic factors. Acquired drivers include autoantibodies, called C3 nephritic factors (C3Nef), that stabilize the C3 convertase [64]. In the dense deposit disease subtype of C3G, 78% of patients express C3Nefs [65]. Similarly, genetic drivers of C3G commonly lie in the *C3* and *CFB* as well as *CFH* loci [66]. AMD is the leading cause of visual impairment in developed countries. In the early stage of disease development, extracellular deposits of lipids and proteins accumulate between the retinal pigment epithelium and the Bruch’s membrane. Later stages of the disease result in extensive damage of the retinal pigment epithelium and eventually loss of vision [67]. An analysis of genetic data from more than 17,000 cases indicated an association between disease and mutations near the genes encoding FH, FI, FB, C2 and C3 [68].

The contexts in which the CP of complement is involved are broad, from the homeostatic removal of apoptotic material to induction of heavy inflammation in host tissues. On host cells, tagging by the C1 complex leads to signaling for clearance of the debris. Importantly, this takes place in the absence of inflammation and lysis, due to the action of FI and the regulators, which promptly degrade C3b to iC3b and dissociates the convertases [69]. It is well-established that deficiency in the early CP components leads to autoimmunity and development of autoantibodies against neoepitopes on the surface of apoptotic cells, due to impaired clearance of apoptotic material [70]. Furthermore, autoimmunity caused by CP activation is also observed in the acute diseases ischemia reperfusion injury, sepsis, antibody induced hemolytic anemia, antibody mediated rejection and cold agglutinin disease [71,72,73,74,75].

In recent years, it has been established that the classical pathway has a well-defined role in the developmental process of synaptic pruning [76], a process required in the developing brain to establish proper synaptic connectivity for a functioning adult brain. Less active synapses are pruned away and one of the mechanisms guiding this process is activation of the complement system through the classical pathway with C4 cleavage, progressing further into the alternative pathway resulting in C3 cleavage [77,78]. In the neurodevelopmental disorder schizophrenia, genome wide association studies have shown a correlation of a C4 isotype overexpression with disease development, and a polymorphism in the central nervous system (CNS) specific functional homologue of CR1, CSMD1 was identified as a risk factor [79,80,81]. Most recently, C4 overexpression was linked to hypo-connectivity in the prefrontal cortex, and schizophrenia-like symptoms in mice [82], and the current hypothesis for schizophrenia pathogenesis involves aberrant complement mediated synaptic pruning. Synaptic pruning onset was also documented during West Nile virus infection and in the neurodegenerative diseases Alzheimer’s, Parkinson’s, multiple sclerosis, frontotemporal dementia and spinal muscular atrophy [83,84,85,86]. Hence, the normal function of the CP during development can be aberrantly reactivated with devastating consequences in adulthood [78,83,84,85,87] and for this reason the CP proteins represent promising therapeutic targets for treatment of neurological diseases with different etiology [86]. 

## 2. The Complement-Targeting Nanobodies

### 2.1. Complement-Specific Nanobodies for Therapy, Research and Diagnostics

The examples presented above underscore the complement system as a therapeutic target, and for more than a decade, there has been significant efforts with respect to developing therapeutic molecules for control of the complement cascade based on e.g. antibodies, pathogen complement evasion proteins, small molecules, peptides, and gene therapy [88,89]. In addition, new reagents for capture and detection of complement proteins may improve diagnostics for the evaluation of future complement therapeutics, which currently relies on the assessment of complement levels, functional assessment of specific pathways and detection of complement degradation products [90]. The involvement of complement in diseases such as Alzheimer’s disease and schizophrenia remains enigmatic. Administration of pathway-specific inhibitors to the brain parenchyma in murine models may allow an accurate mapping of reactions and interactions in the complement system underlying neurological diseases, but delivery across the blood-brain barrier of antibodies and other complement therapeutics represents a severe challenge.

Single domain antibodies derived from heavy chain only antibodies found in animals such as llamas (*llama glama*) called nanobodies [91], are attractive but also largely unexplored modalities for functional analysis and therapeutic control of the complement system. Protocols for selection and development of nanobodies are well established [92]. In the past years, we have developed a portfolio of high-affinity nanobodies specific for proteins in the complement system. A llama is immunized with four injections of 500 µg of the complement antigen, in intervals of three weeks. We isolate the peripheral blood lymphocytes from the immunized llama, extract the RNA, and generate cDNA. This allows us to construct phage display libraries and select nanobodies using in vitro panning. In the following, we provide an overview of complement targeting nanobodies developed by us and others.

### 2.2. The Complement Deposition Assays and Hemolytic Assays

We and others have evaluated the activity of complement inhibitors, including nanobodies, using multiple complement deposition assays (Figure 2A). The typical assay starts by coating the surface with the complement activator of choice, which depends on the pathway investigated. For the CP, the wells are coated with heat aggregated normal human IgG [93], for the LP they are coated with mannan [94,95], a cell wall polysaccharide found in yeast. In assays for AP activity, the wells are coated with zymosan [96], a glucan formed by beta-1,3-glycosidic linkages, simulating the coating of a bacterial surface. The source of complement is normal human serum, which is added in each well at a given dilution depending on the dynamic range of the assay, together with a titration of the nanobody under investigation. Activation leading to deposition is carried out at 37 °C for 90 min, after which the plate is washed and the deposition of complement fragments is detected by addition of a polyclonal antibody specific for C3c or C4c, conjugated to biotin. The readout of the assay are the fluorescence counts generated by europium-conjugated streptavidin. If the counts number is comparable to the counts obtained in the absence of the nanobody, the nanobody is non-inhibitory, while if they are lower, the nanobody has inhibitory activity (Figure 2A, right). Another useful assay for evaluation of complement activation monitors hemolysis of sheep red blood cells sensitized with IgG or IgM antibodies. The red blood cells are lysed only if the MAC complex is assembled on their surface, which is only possible if the CP C5 convertase or the AP C5 convertase formed through the AP amplification loop cleave C5 (Figure 2B). Absence of hemolysis confirms that the nanobody inhibits assembly of the MAC directly or indirectly by inhibiting one of the upstream pathways.

### 2.3. C1qNb75 Inhibits Initiation of the Classical Pathway

The C1q specific nanobody C1qNb75 was selected from a phage display library generated after immunization of a llama with human C1q [97]. C1qNb75 was chosen among a panel of C1q specific nanobodies based on its ability to inhibit CP activation by IgG when analyzed in C4 and C3 fragment deposition assays (Figure 3A). C1qNb75 binds with high affinity (dissociation constant, K_D_=0.3 nM, Table 1) to the isolated recombinant globular domain of C1q suggesting that it may prevent CP activation by occluding the IgG binding site on C1q. Indeed, the nanobody directly blocks C1q binding to IgG coated surfaces and to IgG-antigen immune complexes in a concentration dependent manner. Interestingly, in the C3 deposition assay a 6-fold molar ratio of C1qNb75 to C1q was necessary to fully inhibit CP activation, indicating that all IgG Fc binding sites on the hexameric C1q need to be occupied by the nanobody. We also validated the CP inhibitory activity of the C1qNb75 nanobody with respect to the propagation to the terminal pathway in hemolysis assays. Sheep red blood cells were either coated with IgG or IgM, and release of hemoglobin was measured in the presence of C1qNb75. In accordance with the inhibition of C3 deposition, C1qNb75 also prevented hemolysis (Figure 3B). Finally, to obtain the structural basis for C1qNb75 inhibition of the CP, we determined the structure of the nanobody in complex with the C1q globular domain [97]. The resulting crystal structure determined at 2.2 Å resolution demonstrated that C1qNb75 binds to chain B and chain C of C1q and revealed that C1qNb75 acts by imposing steric hindrance on the C1q globular domain such that the interaction with the Fc of IgG and IgM is prevented (Figure 3C–E).

In conclusion, the data show that C1qNb75 is a potent inhibitor of antibody-mediated CP activation. C1qNb75 may thus represent a candidate for further development of therapeutic molecules for treatment of diseases where IgG/IgM autoantibodies mediate excessive complement activation. C1q binds a multitude of ligands besides IgG and IgM, ranging from patterns on synapses during development and disease, to acute phase proteins like CRP and pentraxin 3. We do not currently know which C1q interactions besides those formed with IgG and IgM the nanobody inhibits. C1qNb75 may however be used as a convenient tool to probe the regions on C1q involved in recognition of these non-antibody ligands. Unfortunately, the nanobody is not cross reactive towards murine C1q but may react with C1q from non-human primates [97]. Nonetheless, C1qNb75 may still be useful in transgenic murine disease models where the human genes for the three C1q subunit fully or partially replaces the murine C1q genes. Such a chimeric C1q was recently used to evaluate the in vivo potential of a bispecific antibody allowing the targeting of human C1q to infectious microorganisms and cancer cells [103].

### 2.4. hC4Nb8 Inhibits In Vivo Assembly of the CP and LP C3 Convertase

Moving to the next step of the classical pathway, a target which is shared with the LP is the proconvertase C4bC2. Inhibition at this stage will not affect the C1q functions independent of CP downstream activation [104] but will attenuate both the CP and LP cascades before C3 cleavage takes place. Hence, an inhibitor of the CP C3 convertase is predicted to tune down the effects of complement activation in diseases linked to CP and LP over-activation, while not affecting basal AP activity (Figure 1). The nanobody hC4Nb8 was selected against human C4b by phage display after immunization of a llama with human C4b. To screen for the presence of complement inhibitors, CP deposition assays were performed and evaluated at the level of C4 and C3 fragment deposition in the presence of the nanobody [98]. In these assays, hC4Nb8 inhibited C3 fragment deposition in a dose dependent manner without influencing C4 deposition (Figure 4A,B). After preliminary experiments showing that the nanobody inhibits C4b2 formation in a size exclusion chromatography assay, hC4Nb8 in complex with C4b was crystallized and the structure of the complex was determined at 3.3 Å resolution [98]. Both complement C3 and C4 comprise eight macroglobulin-like (MG) domains, a C1r/C1s, Uegf, Bmp1 (CUB) domain, a thioester domain (TE), an anaphylatoxin (ANA) domain and a C-terminal C345c domain (Figure 4C). Proteolytic activation of C3 and C4 to C3b and C4b, respectively, releases the ANA domain and leads to a dramatic conformational change [15,16,26,27] that allows the thioester in nascent C4b and C3b to react with a nucleophile on the activator. The atomic model of C4b in complex with hC4Nb8 revealed that the epitope of the nanobody encompasses residues in an N-terminal region of the α’-chain of C4b, along with some residues on the MG6 and MG7 domains (Figure 4D,F). This is a neoepitope generated upon C4 cleavage and is part of the binding site for C2 [105]. In agreement with this, the nanobody binds with a dissociation constant of 16 pM to C4b, a 10^5^-fold higher affinity compared to native C4 (2 µM, Table 1). The steric hindrance imposed by hC4Nb8 on C2 binding was confirmed by comparison of the C4b2 proconvertase model obtained from small angle X-ray scattering (SAXS) data [106] (Figure 4E) and a 3D reconstruction of C4b2 obtained by electron microscopy with the crystal structure of the C4b:hC4Nb8 complex [98]. However, hC4Nb8 also inhibited CR1 mediated FI cleavage of C4b, in agreement with the predicted steric overlap of hC4Nb8 with the putative binding site for CR1 on C4b [44,98,107].

The atomic structure of the C4b:hC4Nb8 complex in combination with the functional data provided a thorough mechanistic basis for understanding how complement inhibition is exerted by hC4Nb8. To test whether hC4Nb8 could control CP activation in vivo, we took advantage of a transgenic hC4 knock-in mouse model and performed a passive immunization experiment. The nanobody was injected subcutaneously along with the high molecular weight red fluorescent antigen phycoerythrin, allowing local CP activation, resulting in C4 and C3 deposition on immune complexes formed between antibodies and the phycoerythrin antigen. Such iC3b coated immune complexes are transported to lymph nodes and docked on follicular dendritic cells for long term antigen presentation important in the process of B cell germinal center formation, and generation of antibodies [108]. By immunostaining of the popliteal lymph nodes, we evaluated the amount of phycoerythrin transported to the B cell follicles in the presence of hC4Nb8 or a control nanobody. When hC4Nb8 was injected, the amount of antigen transported to the lymph nodes was significantly lower, as evaluated by the reduced fluorescence level of phycoerythrin colocalizing with follicular dendritic cells. Though hC4Nb8 was evaluated in a transgenic mouse model expressing human C4 alleles, the nanobody also binds to mouse C4b with an affinity of 0.2 nM (Table 1), holding promise for its application in mouse models of disease linked to CP and LP overactivation. In the context of the CNS, hC4Nb8 was also shown to inhibit C3 deposition on neuronal cell cultures exposed to normal human serum as the source of complement [98]. In summary, the hC4Nb8 stands out as a powerful inhibitor of the complement cascade with an ultrahigh affinity for a neoepitope on C4b capable of interfering with complement activation through both the LP and CP. It is so far the only complement specific nanobody for which in vivo efficacy has been demonstrated, and it therefore appears useful for future work in animal disease models. Remarkably, despite very comprehensive efforts with respect to the development of complement inhibitors [88,89], it represents the only antibody based C4b specific inhibitor described today.

### 2.5. The hC3Nb1 Is an Inhibitor of AP Proconvertase Assembly

Complement component C3 constitutes the point of convergence between the three pathways of the complement system, and C3 is a common target for complete inhibition of the complement cascade [88,89]. To date, we have described the three C3 specific nanobodies hC3Nb1 [99], hC3Nb2 [100] and hC3Nb3 [101]. Both hC3Nb1 and hC3Nb3 prevent the zymogen FB from binding to C3b by direct competition and inhibit C3 degradation through the alternative pathway. In contrast, hC3Nb2 functions by preventing the substrate C3 from binding to either of the two C3 convertases, C4b2a and C3bBb. 

We selected the first C3-specific nanobody, hC3Nb1, against human C3b, but the nanobody binds both native C3 and C3b strongly, with sub-nanomolar dissociation constants (Table 1). In complement deposition assays, the nanobody inhibits the progression of the AP mediated C3 fragment deposition (Figure 5A). In contrast, the nanobody did not inhibit C3 fragment deposition through the CP (Figure 5B) or CP mediated hemolysis of sheep erythrocytes (Figure 5C). As described above, C3 shares the overall domain structure with C4 (Figure 5D,E) and the crystal structures revealed that the hC3Nb1 nanobody binds at the interface between the MG6 and MG7 domains of native C3 and C3b [99,109] (Figure 5F). By comparing the structure of hC3Nb1 in complex with C3b to the structure of the proconvertase C3bB (Figure 5G) [30], we predicted that the nanobody inhibits the assembly of the alternative pathway C3 proconvertase through a steric clash with a glycan on FB. Through size exclusion chromatography, we showed that the presence of the nanobody indeed prevented C3bB assembly. Furthermore, when the glycan was eliminated from FB, hC3Nb1 no longer inhibited the assembly of the proconvertase. 

Since hC3Nb1 was observed to bind tightly to both native C3 and C3b, it was also investigated whether it may act at the level of C3 as a substrate for the C3 convertases. While the crystal structures of the both the AP C3 convertase and the properdin-stabilized AP C3 convertase are known [29,39,110] (Figure 5H), structures of the substrate-bound convertases are not available, which complicates the prediction of how a C3 binding nanobody influences substrate recognition. However, we earlier proposed a general model for substrate-convertase interactions based on structural data [111]. This model states that the C3 convertases primarily recognize the substrate C3 through interactions with the C3b or C4b non-catalytic subunits, respectively, to position the scissile bond in the C3 substrate such that it can be cleaved by the catalytic subunits Bb and C2a in the AP and CP C3 convertase, respectively (Figure 5I). Specifically, the MG4 and MG5 domains in the C3 substrate are believed to interact with the MG4 and MG5 in the convertase C3b/C4b while the C3 substrate MG7 domain is suggested to be recognized by the MG6 and MG7 domains in the convertase C3b/C4b.

In the light of this model, the location of hC3Nb1 on C3 suggested that the nanobody and C3 compete for binding to C3 as their binding sites overlap (Figure 5F vs Figure 5I). However, in the CP complement deposition assay, addition of hC3Nb1 conferred a non-significant effect, suggesting that the competition between hC3Nb1 and the CP C3 convertase is insufficient for inhibition of the CP C3 convertase or that the convertase-substrate model is imperfect. However, in accordance with this model hC3Nb1 did reduce the cleavage of C3 by the fluid phase AP C3 convertase CVF-Bb formed by cobra venom factor (CVF) and factor B [99]. CVF is a convenient reagent for analyzing C3 binding nanobodies as it allows uncoupling of the functions of C3 as a substrate and C3b as the non-catalytic subunit in the AP C3 convertase. These data obtained with the CP C3 convertase and CVF-Bb suggest that preventing the C3 substrate from binding to C3 convertases contributes modestly to the inhibition mode of hC3Nb1.

Importantly, it was also observed that hC3Nb1 interferes with the FH mediated cleavage of C3b by FI [99], which agrees with a predicted overlap between hC3Nb1 and the CCP1 domain of the FH cofactor (Figure 5J). The outcome of AP inhibition with hC3Nb1 in an in vivo setting will hence be the net result of these two opposing effects. On one hand, the nanobody will very efficiently inhibit the C3 turnover through the AP. On the other hand, the nanobody inhibits the endogenous regulation of the deposited C3b generated by the CP C3 convertase. As murine models are extensively used to study pathogenesis of complement associated diseases, we also investigated whether hC3Nb1 inhibits C3 fragment deposition in murine serum. Indeed, hC3Nb1 does inhibit the AP in an assay conducted with murine serum, making it suitable for evaluation of AP inhibition in murine disease models. Lastly, our crystal structure allowed us to localize specific residues of the interaction interface between the nanobody and C3b (Figure 5K,L). This analysis allowed us to construct the inactive mutant of hC3Nb1 (W102A), which does not inhibit the AP (Figure 5A) and hC3Nb1 W102A is hence a perfect negative control for hC3Nb1. Overall, hC3Nb1 represents a high-affinity and cross-reactive C3 specific inhibitor of the alternative pathway mainly acting by preventing proconvertase assembly.

### 2.6. hC3Nb2 Prevents Substrate Binding to C3 Convertases

The second C3 specific nanobody we characterized was hC3Nb2 [100], which was also selected using human C3b. C3 deposition assays revealed that the nanobody inhibits C3 deposition through all three pathways (Figure 5A,B), and furthermore inhibited C3 cleavage by the CVFBb C3 convertase. This suggested that hC3Nb2 acts on the substrate level to prevent the substrate from binding to the convertase and thereby inhibits C3 degradation in contrast to hC3Nb1 that primarily acts by preventing proconvertase assembly. Deposition assays conducted with murine serum mirrored the results obtained in human serum showing that the nanobody inhibits both LP and AP mediated C3 fragment deposition in murine serum [100]. Accordingly, bio-layer interferometry analysis revealed similar dissociation constants of the nanobody for binding to murine and human C3b (Table 1) strengthening the notion that hC3Nb2 is a cross-reactive complement inhibitor. Using negative stain electron microscopy, we mapped the epitope of the nanobody to the MG3 and MG4 domains (Figure 5F) of C3c, a stable degradation product of C3b. This binding site exhibits structural conservation amongst multiple functional states of C3 including native C3, C3(H_2_O), C3b, iC3b and the C3c fragment used for mapping. This was confirmed by affinity measurements, where hC3Nb2 bound the C3(H_2_O) analogue C3 methylamine (C3MA), C3b and native C3 with dissociation constants in the low nanomolar range (Table 1). Since the binding site is shared between C3c and native C3, we could construct an overall model of the C3-hC3Nb2 complex. Comparison of this with our general model of the convertase-substrate complexes [111,112] (Figure 5I) confirmed that hC3Nb2 interferes with C3 substrate recognition by both C3 convertases C4b2a and C3bBb.

Lastly, we compared a model of the C3b-hC3Nb2 complex to the structure of the ternary complex between C3b, FI and FH [107] (Figure 5J). This comparison indicated that the hC3Nb2 epitope lies far from the binding sites of FI and FH, and predicted that the nanobody in contrast to hC3Nb1 does not interfere with endogenous regulation of the deposited C3b. Accordingly; a fluid phase assay revealed no effect of hC3Nb2 on the FH mediated cleavage of C3b by FI [100]. The cross-reactive hC3Nb2 nanobody thus inhibits all three pathways of the complement system, while it allows the degradation of C3b and it hence constitutes a potent inhibitor of the system. One disadvantage of the nanobody is however that native C3 normally present at seven μM in serum must be saturated, implying that hC3Nb2 may need to be administrated such that its concentration remains at 0.1 mg/ml if systemic C3 deposition is to be completely eliminated.

### 2.7. hC3Nb3 Provides Insight into the Role of C3b in C5 Convertases

The third C3-specific nanobody, hC3Nb3, was selected against the C345c domain of murine C3 from a library obtained from a llama immunized with human C3b. This selection strategy favored the selection of cross-reactive nanobodies that bind both human and murine C3. We demonstrated that hC3Nb3 inhibits C3 deposition through the alternative pathway although less efficiently in murine serum as compared to human serum, where it efficiently blocks C3 fragment deposition (Figure 5A). Next, the complex between the nanobody and the antigen C345c domain of murine C3 was crystallized, and the structure of the complex was determined to a maximum resolution of 1.5 Å [101]. The structure revealed, that the nanobody grasps around the C-terminal helix of C3b, through the formation of a concave binding site on the nanobody. Interestingly, the nanobody binds in a sideways manner and forms a binding site that involves extensive interactions between the framework region and the C-terminal helix of C3b (Figure 5M,N). Within the AP C3 proconvertase, FB interacts with the C345c domain of C3b, and by comparison of the crystal structure of the hC3Nb3:C345c complex with the structure of C3bB [30] (Figure 5G), we predicted that the nanobody would prevent the assembly of the alternative pathway C3 convertase, which was confirmed by size exclusion chromatography. In contrast, our structure also predicted that hC3Nb3 does not prevent substrate C3 binding to the CP C3 convertase or by the CVFBb C3 convertase, which was also confirmed experimentally (Figure 5B).

Based on the data presented above, hC3Nb3 appeared to be a highly specific AP inhibitor similar to hC3Nb1. However, deeper functional analyses showed that there are important differences between the two nanobodies. When projected onto C3b, the hC3Nb3 epitope overlaps with the binding sites of AP regulators properdin (Figure 5H) and factor I (Figure 5J). Accordingly, using bio-layer interferometry, we could confirm that in contrast to hC3Nb1, hC3Nb3 inhibits the interaction between C3b and properdin. Hence, in addition to preventing C3b binding to factor B, hC3Nb3 also prevents properdin from stabilizing the convertase. This further potentiates the inhibitory effect of the nanobody with respect to the AP as FP deficiency and depletion result in highly attenuated AP activity [110]. With respect to FI degradation of C3b to iC3b, the in vitro cleavage assay somewhat surprisingly showed that hC3Nb3 only delayed the FH mediated cleavage of C3b by FI rather than completely preventing it. This observation suggests that the interaction between FI and the C3b C345c domain defined in the structure of the C3b:FH:FI complex [107] is not essential for FI recognition of the C3b:FH complex.

An additional intriguing observation was that hC3Nb3 inhibited the formation of the C5b-9 complex upon activation of both CP and LP, which was unexpected since hC3Nb3 did not affect C3 deposition by the CP C3 convertase [97] (Figure 5B). To gain further insight into the apparent inhibition of the CP/LP C5 convertase, the effect of the nanobody on CP mediated lysis of sheep erythrocytes was tested. Similar to the C5b-9 deposition assays, hC3Nb3 efficiently reduced CP driven hemolysis [101] (Figure 5C). To exclude that this effect was due to residual AP activity, we also performed experiments in FB depleted serum that does not allow the assembly of the AP C3 convertase. This assay similarly showed a reduced hemolysis of sheep erythrocytes in the presence of hC3Nb3. As described above, buildup of the density of C3b on the activating surface induces a specificity shift in the two C3 convertases to function as C5 convertases. One explanation for the effect of hC3Nb3 on the CP mediated hemolysis is hence, that the nanobody prevents the function of these auxiliary C3b molecules in the C5 convertase. As the AP C3 convertase does not form efficiently in the presence of hC3Nb3, this effect could only be analyzed for the CP C5 convertase. Whether the nanobody directly inhibits the C5 substrate recognition or rather the assembly of the CP C5 convertase is difficult to determine as the exact role of C3b in the CP C5 convertase remains unsettled [112]. In summary, hC3Nb3 is a high-affinity AP inhibitor preventing assembly of the proconvertase without interfering strongly with FI degradation of C3b. Through its binding to C3b, hC3Nb3 also interferes with the substrate cleavage by the CP C5 convertase through a yet unknown mechanism.

### 2.8. The Noninhibitory Properdin-Specific hFPNb1

Whereas the C1q, C4 and C3 specific nanobodies described above are inhibitory and therefore candidates for complement therapeutics, the properdin specific nanobody hFPNb1 offers a demonstration of how non-inhibitory nanobodies can support structure-function studies of complex molecular pathways like the complement cascade. As alluded to above, properdin circulates in three oligomeric states in circulation and adopts very extended structures [113]. To tackle its structure determination, Pedersen et al. generated a monomeric version of properdin, named FPc [109]. In addition, Pedersen et al. generated a library and selected nanobodies against properdin, in order to isolate nanobody binders that could promote structure determination of the complex between the monomeric properdin and the AP C3 convertase C3bBb. The hFPNb1 nanobody was crystallized in complex with C3bBb:FPc and the staphylococcal complement inhibitor SCIN at 6.2 Å resolution [39]. The structure revealed its epitope located on FPc TSR4, far from C3b and Bb, in accordance with absence of AP inhibitory activity of this nanobody. The study underlines the well-known potential of nanobodies as crystallization chaperones that can trap macromolecules in a specific conformation, enabling crystallization of challenging targets including membrane proteins and their complexes.

### 2.9. Vsig4-Specific Nanobodies

In addition to the nanobodies described above, a nanobody targeting the Vsig4 receptor (also known as complement receptor Ig, (CRIg) has also been characterized [114]. Besides being involved in regulation of T cell immunity, Vsig4 is also a complement receptor expressed exclusively on tissue resident macrophages, where it mediates the clearance of circulating complement-opsonized cells, through recognition of the MG4 and MG5 domains of C3b and iC3b [47,115]. The ectodomain of Vsig4 is a possible therapeutic molecule that through binding to C3b and iC3b may control diseases associated to excessive CP and AP activation [116]. Vsig4 specific nanobodies were selected from a library obtained after immunization of an alpaca with the extracellular domain of murine Vsig4. The nanobody Nb119 binds to the binding site of the C3b MG4 and MG5 domains on Vsig4, thereby preventing the endogenous interaction [114]. The nanobody is cross reactive with human Vsig4, albeit with a 250-fold lower affinity. The crystal structures of the complexes of the Nb119 with human and murine Vsig4 demonstrated that the decreased affinity for hVsig4 is due to changes in CDR3 of Nb119. Single residue mutations guided by the available atomic models could be introduced for improving the Nb119 affinity toward hVsig4 [114].

## 3. Perspectives

### 3.1. Complement-Specific Nanobodies as Tools in Diagnostics and Research

Due to their single domain nature, nanobodies are relatively easy to engineer and multiple examples of insertion of site-specific modifications or fusion proteins have been described in the literature. These include addition of fluorophores and radioligands, and fusion to Fc-fragments or fluorescent proteins. For live imaging, nanobodies are superior to monoclonal antibodies, due to their shorter circulation time, which results in a decreased background signal. An interesting feature of the hC3Nb3 nanobody is its very strong and sodium dodecyl sulfate (SDS) resistant interaction with the C345c domain of C3. We frequently observed the formation of SDS polyacrylamide gel electrophoresis resistant complexes between the nanobody and the various functional states of C3, which was used to assay the relative amount of the individual C3 degradation products in samples [101]. This also led us to explore other applications of the nanobody. Through fusion to an IgY Fc fragment, we produced the nanobody into an antibody format and directly conjugated a fluorophore to the fusion protein. Using this hC3Nb3-IgY Fc fusion protein, we performed immunohistochemistry of C3 deposited on in vitro differentiated neurons. This staining revealed C3 deposition on a subset of the neurites [101]. In an orthogonal approach, C3 fragments on pig erythrocytes were quantitated by flow cytometry using a hC3Nb3-GFP fusion protein as the marker [101]. These properties demonstrate the potential of using the hC3Nb3 nanobody in diagnostic applications quantitating C3 deposition. An additional example is offered by the Vsig4 specific nanobody Nb119 labeled with ^99m^Tc [102]. This allows imaging of inflamed joints in collagen induced arthritis (CIA) mice, a model that mimics the human rheumatoid arthritis immunologic state [117]. The accumulation of ^99m^Tc-Nb119 in CIA mice joints correlated well with disease severity assigned by the macroscopic arthritis scoring system, but the nanobody also allowed detection of knee inflammation prior to onset of macroscopic symptoms [102]. The results from this study indicate that a similar approach could be followed to diagnose human rheumatoid arthritis even before macroscopic detection of joint inflammation is possible. The authors further characterized the ability of ^99m^Tc-Nb119 to follow the stages of disease development by using STIA mice, where arthritis is temporarily induced by transfer of arthritogenic serum and resolves after 15-30 days [118]. In this way, Zheng et al. showed that ^99m^Tc-Nb119 signal correlates with the stage of disease progression, demonstrating its applicability to monitor disease evolution in a non-invasive manner [119].

Complement specific nanobodies have also demonstrated to be valuable reagents for basic research aiming at analyzing the contribution of the individual pathways to activation of the cascade. Anti-αGal antibodies were thought to inhibit complement on pathogen surfaces, nonetheless these antibodies react with a number of pathogens [120]. Jensen et al. recently utilized hC3Nb1, hC3Nb2, and C1qNb75 to attribute the complement pathways elicited by these antibodies targeting the galactose-α-1,3-galactose carbohydrate. The authors reported that the anti-αGal antibodies elicit complement deposition by activating the classical pathway [121]. This basic research application is an example of how complement inhibitory nanobodies can be used to assign the contribution of the individual pathways to complement activation. Recently, Lausen et al. utilized hC3Nb2 to broadly inhibit the in vitro complement deposition on *Chlamydia trachomatis* and demonstrated complement dependent adherence of the bacteria to B-cells [122].

### 3.2. Nanobody-Driven Activation of Complement

Bispecific monoclonal antibodies that target bacterial surface proteins or B and T cell antigens with one arm and C1q with the other arm were previously investigated as a means of targeting complement to specific cell types. It was demonstrated that such bifunctional antibodies could elicit complement dependent cytotoxicity on different types of cells [103]. The potential of complement targeting by nanobody fusion proteins was explored by Pedersen et al., who directed complement activation to tumor cells by combining hFPNb1 with a nanobody targeting the epidermal growth factor receptor expressed on cancer cells [123]. Although C3 deposition could be detected on the cells targeted, this did not result in complement dependent cytotoxicity [123]. It was not possible to conclude whether this was due to absence of C5 convertase formation or due to a high abundance of complement regulators on the surface of cancer cells. Nevertheless, the demonstration that nanobody driven targeting of properdin supports AP initiation offers a tool to future experiments aiming at exploring the somewhat controversial concept that properdin can act as C3b independent pattern recognition molecule capable of initiating the AP pathway on a surface without prior C3 deposition [35].

### 3.3. Nanobodies as Complement Therapeutics

The complement system is now recognized to be involved in the pathogenesis of numerous diseases and consequently inhibiting complement is gaining increased attention. Currently, the only FDA approved complement specific inhibitors are the anti-C5 monoclonal antibody eculizumab and its derivative ravulizumab (Alexion pharmaceuticals). Both antibodies inhibit C5 cleavage and initiation of the terminal complement pathway. Eculizumab is indicated for treatment of patients with PNH, aHUS and generalized myasthenia gravis while ravulizumab is approved for treatment of patients with PNH and aHUS. However, with the general acceptance of complement proteins as validated therapeutic targets, many other complement modulating compounds are now in development [88,89]. These include monoclonal antibodies, peptides and small molecules, but complement specific nanobodies thus far represent an underexplored modality for complement inhibition. While mAbs are widely applied as therapeutics only one nanobody, caplacizumab, a dimeric nanobody against Von Willebrand factor, is clinically approved [124]. Both nanobodies and mAbs bind with high affinity and specificity to their antigen, but nanobodies lack an Fc-fragment and consequently do not engage Fc-receptors and C1q. Due to their large size, mAbs often target surface exposed and flat epitopes, while nanobodies can penetrate cavities and often have concave three-dimensional epitopes. Hence, epitopes targeted by nanobodies often differ by the ones of mAbs, which may allow complementary applications, for example in targeting the antigen in autoantibody-mediated conditions. In addition, nanobodies have improved tissue penetration due to their small size and may access inner layers of tumors and brain tissues. In comparison to small molecules, which require several rounds of screening and rational design to achieve high affinity and time demanding synthesis steps, selection of the nanobodies from libraries generated by immunization of a llama allows rapid isolation of high affinity binders, significantly reducing their development time. Furthermore, nanobodies display high specificity for their targets, which is not always achieved by small molecule therapeutics.

For these reasons, we foresee an increased interest in developing complement specific nanobodies due to their unique characteristic of binding with high affinity and specificity to their target antigen combined with their single domain structure, allowing easy protein engineering. The residual immunogenicity of nanobodies in connection to their animal origin can be overcome by humanization of the residues in their framework regions [125]. The small size of nanobodies leads to rapid renal clearance upon intravenous administration. Compared to IgG1 which has a typical half-life of 21 days in humans, nanobodies have a half-life of 1-2 h in mice serum [126]. The low half-life of complement nanobodies may however be extended by the presence of their antigens in blood and contribute to improved circulation times (target-mediated circulation). Several strategies have been investigated to overcome the short serum half-life, including oligomerization of the nanobodies, addition of polyethylene glycol, fusion to an IgG framework, conjugation to albumin binding peptides, formation of nanobody-liposomes, nanobody-micelles or nanobody-albumin nanoparticles [127,128].

One should carefully consider where nanobodies would provide a benefit compared to mAbs. For complement targeting nanobodies it would often be optimal to have them acting at the site of complement activation, to avoid systemic inhibition of the beneficial functions of complement. The potential of site specific delivery of a complement inhibitor was convincingly demonstrated with the protein based complement inhibitor B4Crry in an mouse model of stroke [129]. Nanobodies may be preferred modalities for tissue targeted complement inhibition as they are easily fused to other targeting moieties with minimal protein engineering. The high renal uptake of nanobodies could be exploited to localize the inhibitors to renal tissue in kidney diseases where complement activation is known to cause injury, as in C3 glomerulopathy [64]. Likewise, inhibitory nanobodies could be targeted to aquaporin-4 to provide local complement inhibition and ameliorate the complement activating effect of autoantibodies in neuromyelitis optica [130].

An exciting alternative to systemic administration is local delivery of complement specific nanobodies by infection with adeno-associated virus (AAV) vectors. This could enable delivery to tissues, such as the brain, where repeated administration by other means would not be possible. The role of complement in several CNS diseases is at the centre of a fast growing field of research and nanobodies bear unexplored potential to date as especially useful tools for complement inhibition in this context [86]. A recent study delivered proof of principle with respect to AAV driven delivery of a small protein based complement inhibitor in an animal model of demyelinating disease [131].

## 4. Conclusions

We have presented seven nanobodies that bind different complement components and outlined the biochemical and structural characterization of their interaction with their respective antigens. Besides the inhibitory activity of the nanobodies described in the first part, we discuss potential applications of the nanobodies for detecting complement deposition on cells or for bioimaging of joint inflammation. As basic research tools, nanobodies were shown to mediate targeting of complement to a surface, and to allow unambiguous assignment of the complement activating ability of anti-αGal IgGs. These are a few examples of the opportunities offered by the complement specific nanobodies presented. Further research could identify their potential in transient complement inhibition instead of complement gene knockout in animal models, thus avoiding associated side effects, as development of SLE and auto-reactive B cells for C1q KO and C4 KO mice respectively [75,132]. The complement inhibitory nanobodies may also be administrated in the central nervous system in animal models of neurodegenerative diseases to evaluate the effect of inhibiting especially the C3 convertases and their potential for future therapeutic applications.

## Figures and Tables

**Figure 1 biomolecules-11-00298-f001:**
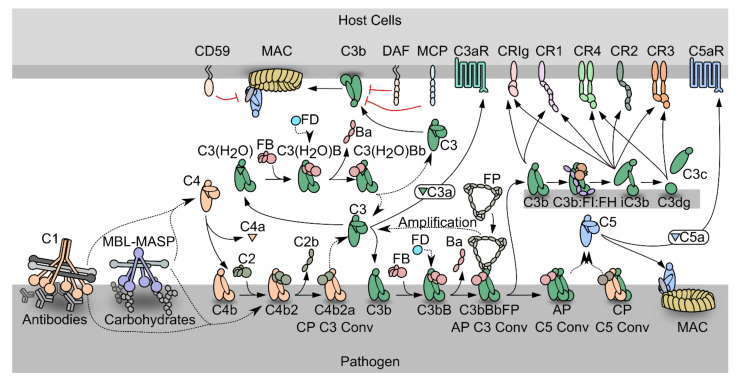
Schematic representation of the complement system. PRM-associated proteases activate upon binding of the PRMs to specific activators. The active proteases cleave C4, which undergoes a conformational change and covalently binds the activator. The zymogen C2 binds the opsonizing C4b, forming the C3 proconvertase. The active C3 convertase cleaves C3, which liberates the C3a fragment from C3b that covalently attaches to the activator. Factor B (FB) may bind the C3b, and upon activation by Factor D, the AP C3 convertase appears, that is stabilized by properdin (FP). C3 may also undergo spontaneous hydrolysis forming C3(H2O), which is a functional homologue of C3b and allows the assembly of a fluid phase C3 convertase. When the C3b concentration reaches a threshold density on the activator, the C3 convertases shifts specificity to C5. The resulting C5 convertases cleave C5 and the C5b forms the starting point for assembly of the membrane attack complex that may perforate the cell membrane. C3b may also undergo degradation by factor I assisted by cofactors. The degradation of C3b opens for interactions with complement receptors.

**Figure 2 biomolecules-11-00298-f002:**
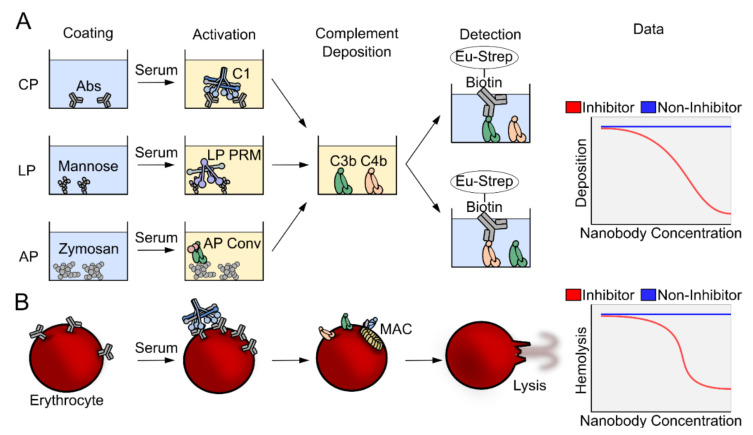
Overview of complement assays. (**A**) In ELISA based complement deposition assays, pathway-specific molecular patterns activate either the CP, the LP or the AP. The covalent binding of complement components allows thorough washing and subsequent readout of either C4 or C3 deposition. (**B**) CP activation can be provoked on sheep erythrocytes using specific antibodies. The complement system may progress to the terminal pathway and the assembly of the membrane attack complex that lyses the erythrocytes. The amount of hemoglobin released can be measured by spectrophotometry at 405 nm for quantification of the fraction of lysed cells.

**Figure 3 biomolecules-11-00298-f003:**
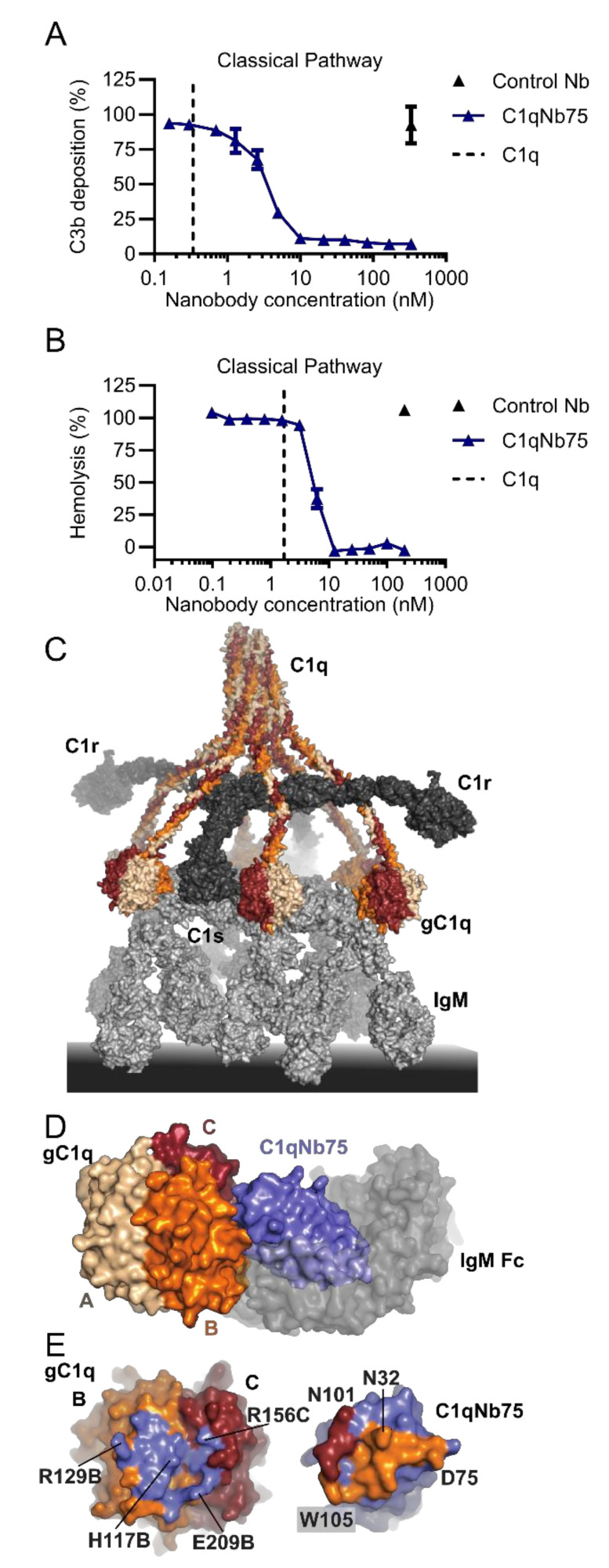
Structure and function C1qNb75. (**A**) Effect of C1qNb75 or control nanobody (Nb) on C3 fragment deposition in 0.2% NHS. The CP was activated by heat aggregated IgG. (**B**) Hemolysis of IgM coated sheep erythrocytes in the presence of C1qNb75 or control Nb in 1% NHS. The putative C1q concentration in the assay is shown as a dashed line. (**C**) Structural model of the C1 complex bound to IgM based on [12]. The three chains of C1q is shown in wheat, orange and dark red. C1r and C1s in dark grey and IgM in light grey. (**D**) Structure of the globular domain of C1q (gC1q) in complex with C1qNb75 (blue) (protein data bank (PDB) entry 6Z6V). IgM-Fc (semi-transparent grey surface) is shown to illustrate the steric clash between C1qNb75 and IgM. Adapted from (man). (**E**) Epitope of C1qNb75 (left) on gC1q shown in blue and paratope on C1qNb75 (right). The paratope on C1qNb75 is colored according to the C1q chain recognized.

**Figure 4 biomolecules-11-00298-f004:**
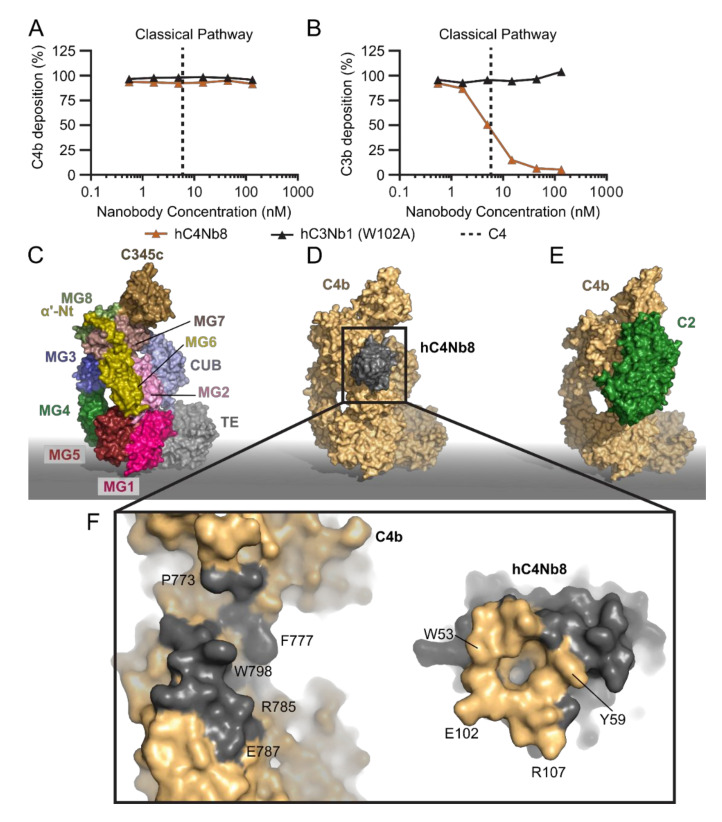
The inhibition mechanism of hC4Nb8. (**A**,**B**) Effect of hC4Nb8 on C3b and C4b deposition in 0.2% NHS where the CP was activated by heat aggregated IgG. hC4Nb8 potently inhibits C3b deposition without interfering with C4b deposition. The inactive hC3Nb1 W102A is included as control. (**C**) Domain structure of C4b (PDB entry 5JTW). Labels indicate the location of the individual domains. (**D**) the C4b:hC4Nb8 crystal structure (PDB entry 6YSQ). (**E**) the C4b2 atomic model based on SAXS data [106] highlights how hC4Nb8 binding to C4b prevents C2 from binding due to pronounced steric hindrance. (**F**) An open book view of the hC4Nb8:C4b interface, revealing formation of a hydrophobic pocket in the hC4Nb8 paratope accommodating C4b Trp798. Important residues mediating the interaction are labeled. In panels (**D**–**F**), C4b is shown in yellow, hC4Nb8 in gray, and C2 in green.

**Figure 5 biomolecules-11-00298-f005:**
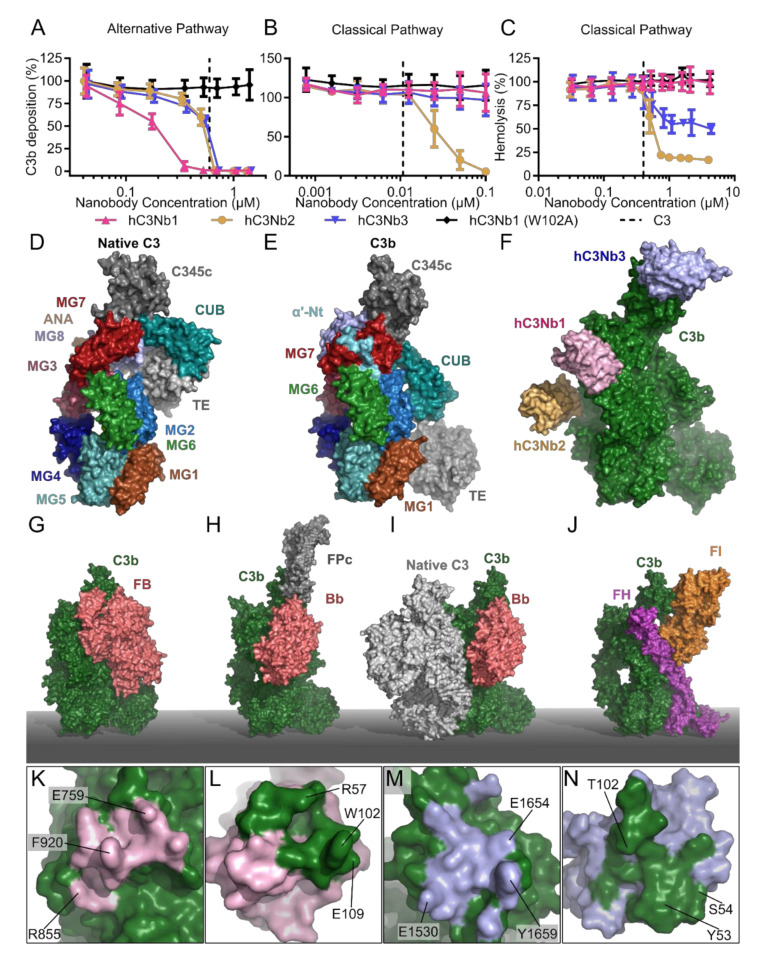
The C3 specific nanobodies. C3 fragment deposition assay elicited by (**A**) the AP in 11% NHS or (**B**) the CP in 0.2% NHS demonstrating the effect of the C3 specific nanobodies: hC3Nb1 (AP inhibitor), hC3Nb2 (CP/LP/AP inhibitor), hC3Nb3 (AP inhibitor) as well as the inactive hC3Nb1 (W102A) mutant. (**C**) Classical pathway mediated hemolysis assay in 7.5% NHS comparing the effects of hC3Nb1, hC3Nb2, hC3Nb3 and hC3Nb1 (W102A). Dashed lines in panels (**A**–**C**) indicate the putative C3 concentration in the assays. Surface representation of (**D**) native C3 (PDB entry 2A73) and (**E**) C3b (PDB entry 5FO7). Labels indicate the location of the individual domains in panels (**D**,**E**). (**F**) The proposed binding site of hC3Nb2 as well as the crystal structure of the hC3Nb3 (PDB entry 6XZU) were docked onto the crystal structure of hC3Nb1 in complex with C3b (PDB entry 6EHG). Crystal structures of (**G**) the AP C3 proconvertase (PDB entry: 2XWJ), and (**H**) the properdin stabilized AP C3 convertase (PDB entry: 6RUR). (**I**) The substrate recognition model of the C3 convertases, where C3 (PDB entry 2A73) and C3bBb (PDB entry: 2WIN) were docked into the structure of the CVF:C5 complex (PDB entry: 3PVM). (**J**) The crystal structure of the C3b:FI:mini-FH complex (PDB entry 5O32). (**K**,**L**) Interaction sites indicated as footprints of (**K**) hC3Nb1 on C3b, (**L**) C3b on hC3Nb1, (**M**) hC3Nb3 on C3b, and (**N**) C3b on hC3Nb3. Labels indicate selected residues in antigen and nanobody of importance for the interaction.

**Table 1 biomolecules-11-00298-t001:** The complement targeting nanobodies presented in this review. The table outlines the inhibitory mechanism of each nanobody, its epitope and antigen. Dissociation constants are given toward different forms of the targeted complement component, and toward the mouse component when available.

Nanobody (Reference)	Mechanism	Epitope	Antigen	Dissociation Constant K_D_ (nM)
C1qNb75 [97]	Prevents C1q binding to Fc of IgG and IgM	gC1q chains B and C	hgC1q	0.3
hC4Nb8 [98]	Blocks CP proconvertase assembly	C4b α’-Nt, MG6 and MG7 domains	hC4	2000
			hC4b	0.016
			mC4b	0.2
hC3Nb1 [99]	Blocks AP proconvertase assembly	C3 and C3b MG6 and MG7 domain	hC3	0.9
			hC3b	0.2
hC3Nb2 [100]	Prevents substrate binding to C3 convertases	C3 and C3b MG3 and MG4 domains	hC3	10
			hC3b	5
			hC3-MA	3
			mC3b	0.6
hC3Nb3 [101]	Blocks AP C3 proconvertase assembly and CP C5 convertase activity	C3 and C3b C345c domain	hC3	3
			hC3b	3
			hC3-MA	6
			mC3b	3
hFPNb1 [39]	Non-inhibitory	FP thrombospondin repeat 4	hFPc	7.3
Nb119 [102]	Blocks C3b and C3c binding to Vsig4 (CRIg)	Vsig4 (CRIg) ectodomain	hVsig4	850
			mVsig4	3.5

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
