# Peer review of "Nanobodies Provide Insight into the Molecular Mechanisms of the Complement Cascade and Offer New Therapeutic Strategies"

_biomolecules, 2021, doi:10.3390/biom11020298_

Round 1
Reviewer 1 Report
Summary: In their article entitled “Nanobodies provide insight into the molecular mechanisms of the complement cascade and offer new therapeutic strategies”, the Andersen lab reviews progress on their recent complement-targeted nanobody studies. The review is timely, well written, and nicely organized. It describes a series of elegant studies that characterize the structure and function of novel nanobodies raised against C1q, C4, C3, properdin, and CRIg. A focus is given to the molecular mechanisms of each nanobody and I find this to be a strength of the review. The idea that these molecules can be used in answering basic research questions, including in addressing the relationship of complement in various disease models, is well-founded and draws parallels to similar concepts from the field of complement evasion. I have a single suggestion for improvement and note a few minor text edits.
Minor suggestions:
- In the Perspectives, I think the review would be improved by having a new section discussing the use of nanobodies as therapeutics. An emphasis on some of the limitations that have prevented their use from taking hold relative to monoclonal antibody therapies could be discussed. As pointed out in reviews such as (Morrison, C. Nat Rev Drug Discov. 2019 Jul;18(7):485-487) momentum for this modality is building, particularly for inhibitors such as those reviewed here where detailed mechanistic studies have been performed that might be taken further advantage of with protein engineering technologies. Some of this is mentioned throughout the review, but a paragraph highlighting the general strengths and limitations of nanobody therapeutics relative to the widespread mAb format would be useful.
Text revisions:
Line 42: chain should be plural
Line 213, 215, 252, 303: The word “lama” should be llama.
Line 642: characteristic needs a space
Author Response
Summary: In their article entitled “Nanobodies provide insight into the molecular mechanisms of the complement cascade and offer new therapeutic strategies”, the Andersen lab reviews progress on their recent complement-targeted nanobody studies. The review is timely, well written, and nicely organized. It describes a series of elegant studies that characterize the structure and function of novel nanobodies raised against C1q, C4, C3, properdin, and CRIg. A focus is given to the molecular mechanisms of each nanobody and I find this to be a strength of the review. The idea that these molecules can be used in answering basic research questions, including in addressing the relationship of complement in various disease models, is well-founded and draws parallels to similar concepts from the field of complement evasion. I have a single suggestion for improvement and note a few minor text edits.
Minor suggestions:
- In the Perspectives, I think the review would be improved by having a new section discussing the use of nanobodies as therapeutics. An emphasis on some of the limitations that have prevented their use from taking hold relative to monoclonal antibody therapies could be discussed. As pointed out in reviews such as (Morrison, C. Nat Rev Drug Discov. 2019 Jul;18(7):485-487) momentum for this modality is building, particularly for inhibitors such as those reviewed here where detailed mechanistic studies have been performed that might be taken further advantage of with protein engineering technologies. Some of this is mentioned throughout the review, but a paragraph highlighting the general strengths and limitations of nanobody therapeutics relative to the widespread mAb format would be useful.
Response:
We have added a new extensive section immediately before the conclusion termed
3.3 Nanobodies as complement therapeutics
We hope this addresses the excellent suggestion made by the reviewer
Text revisions:
Line 42: chain should be plural
Response. Corrected
Line 213, 215, 252, 303: The word “lama” should be llama.
Response. Llama is now used throughout
Line 642: characteristic needs a space
Response. Corrected, now in line 749
Reviewer 2 Report
The review by Zarantonello et al describes the main features of the complement cascade and the findings of the group in isolating nanobodies that are directed against different molecules of the complement system.
Overall, the review is well written, it provides a comprehensive overview of the complement cascade (without repeating previous reviews in this field) and describes the work done by this group in the last few years. Nevertheless, my main comment is that it is not clear what is the main advantage of these nanobodies to further understand the activities of the complement system or the applications in the therapeutic arena. Indeed, the authors do state (lines 639-644) that "These include monoclonal antibodies, peptides and small molecules, but complement specific nanobodies thus far represent an underexplored modality for complement inhibition. We foresee an increased interest in developing complement specific nanobodies due to their unique characteristicof binding with high affinity and specificity to their target antigen combined with their single domain structure, allowing easy protein engineering.". However, I believe that a section that describes and compare the role of "regular" antibodies and small molecules in understanding the activity of the complement system should be more lengthy. In addition, the "perspective" section should be more focused on examples and suggestion of how the use of nanobodies will further increase the knowledge and the therapeutic potential in the field of the complement system, when compared to antibodies and small molecules. In its current version, this section does not support well enough this point.
Author Response
The review by Zarantonello et al describes the main features of the complement cascade and the findings of the group in isolating nanobodies that are directed against different molecules of the complement system.
Overall, the review is well written, it provides a comprehensive overview of the complement cascade (without repeating previous reviews in this field) and describes the work done by this group in the last few years. Nevertheless, my main comment is that it is not clear what is the main advantage of these nanobodies to further understand the activities of the complement system or the applications in the therapeutic arena. Indeed, the authors do state (lines 639-644) that "These include monoclonal antibodies, peptides and small molecules, but complement specific nanobodies thus far represent an underexplored modality for complement inhibition. We foresee an increased interest in developing complement specific nanobodies due to their unique characteristicof binding with high affinity and specificity to their target antigen combined with their single domain structure, allowing easy protein engineering.". However, I believe that a section that describes and compare the role of "regular" antibodies and small molecules in understanding the activity of the complement system should be more lengthy. In addition, the "perspective" section should be more focused on examples and suggestion of how the use of nanobodies will further increase the knowledge and the therapeutic potential in the field of the complement system, when compared to antibodies and small molecules. In its current version, this section does not support well enough this point.
Response. The new section 3.3 includes comparison with conventional antibodies and small molecules and provide examples where nanobodies could be the best modality for a complement inhibitor. We anticipate this new section addresses the overall comment made by the reviewer.